# Degree and site of chromosomal instability define its oncogenic potential

Wilma H.M. Hoevenaar[1], Aniek Janssen [2,5], Ajit I. Quirindongo[1,5], Huiying Ma[3,5], Sjoerd J. Klaasen[1], Antoinette Teixeira[1], Bastiaan van Gerwen[1], Nico Lansu[1], Folkert H.M. Morsink [3], G. Johan A. Offerhaus[3], René H. Medema[4], Geert J.P.L. Kops [1,6✉] & Nannette Jelluma [1,6✉]

Most human cancers are aneuploid, due to a chromosomal instability (CIN) phenotype. Despite being hallmarks of cancer, however, the roles of CIN and aneuploidy in tumor formation have not unequivocally emerged from animal studies and are thus still unclear. Using a conditional mouse model for diverse degrees of CIN, we find that a particular range is sufficient to drive very early onset spontaneous adenoma formation in the intestine. In mice predisposed to intestinal cancer ($Apc^{Min/+}$), moderate CIN causes a remarkable increase in adenoma burden in the entire intestinal tract and especially in the distal colon, which resembles human disease. Strikingly, a higher level of CIN promotes adenoma formation in the distal colon even more than moderate CIN does, but has no effect in the small intestine. Our results thus show that CIN can be potently oncogenic, but that certain levels of CIN can have contrasting effects in distinct tissues.

[1] Oncode Institute, Hubrecht Institute-KNAW and University Medical Center Utrecht, Utrecht, The Netherlands. [2] Center for Molecular Medicine, Section Molecular Cancer Research, University Medical Center Utrecht, Utrecht, The Netherlands. [3] Department of Pathology, University Medical Center Utrecht, Utrecht, The Netherlands. [4] Division of Cell Biology, Netherlands Cancer Institute, Oncode Institute, Amsterdam, The Netherlands. [5] These authors contributed equally: Aniek Janssen, Ajit I. Quirindongo, Huiying Ma. [6] These authors jointly supervised this work: Geert J.P.L. Kops, Nannette Jelluma. ✉email: g.kops@hubrecht.eu; n.jelluma@hubrecht.eu

Aneuploidy—an abnormal number of chromosomes—is a hallmark of human tumors[1]. While during embryogenesis almost all aneuploidies are lethal[2] and aneuploidy levels in normal tissues are very low[3], chromosomal aberrations are observed in 70–90% of solid tumors[4,5]. Aneuploidy is the result of chromosomal instability (CIN): the occasional gain or loss of whole chromosomes during mitosis. CIN can lead to genome destabilization[6–9], and is associated with high intra-tumoral genomic heterogeneity, immune evasion, and promotion of metastases (reviewed in ref. [10]). Furthermore, CIN and aneuploidy are correlated with tumor aggressiveness, therapy resistance and poor patient prognosis[11–19].

Colorectal cancer (CRC) is a very common and deadly cancer type that has a high prevalence of CIN and aneuploidy. CRC is divided in heritable and sporadic types, both mainly consisting of microsatellite unstable (MIN, 15%) and CIN (85%) tumors[20]. CIN correlates with poor patient prognosis: a meta-analysis of 63 studies with a total of 10.126 CRC patients (60% aneuploid tumors, as a proxy for CIN) showed that CIN tumors responded worse than non-CIN tumors to 5-Fluoruracil treatment, and (progression free) survival was lower in CIN patients[14].

Despite the correlations described above, the roles of CIN and aneuploidy in tumor formation are still unclear. Mouse models of CIN have occasionally shown sporadic, spontaneous tumors with very long latency (>12–18 months), and predominantly in spleen and lung[21–29], suggesting CIN is not a potent cancer driver. In mice predisposed to cancer, CIN is either neutral[30–32], promotes tumor formation[23,26,27,30,33–39], or, in some conditions, suppresses it[29,30,32,37]. Comparisons between these studies is however exceedingly difficult due to the use of different oncogenic backgrounds, to differences in tissues that were examined[40], and to the manner and time by which the tissues were exposed to CIN. Moreover, technical limitations often precluded direct measurements of CIN in the relevant tissues, and oncogenic effects in the different models cannot be attributed to distinct CIN levels.

We therefore established a genetic mouse model that allows controlled induction of various degrees of CIN in a tissue-specific manner. With this model, we make direct comparisons and find striking differences in the consequences for tumor formation between the various degrees of CIN: moderate to high CIN levels are sufficient to drive very early intestinal tumor formation, with moderate CIN being the most effective. Moreover, the capacity of similar CIN levels to drive or promote tumor formation is not the same in distinct tissues.

## Results

### An allelic series for graded increases of CIN in vivo.
To enable tissue-specific induction of a range of CIN levels, we created mouse strains carrying a conditional T649A (TA) or D637A (KD; kinase-dead) mutation in the spindle assembly checkpoint kinase *Mps1* (Fig. 1a, b, Supplementary Fig. 1A–C). Similar mutations in human cell lines caused mild or severe CIN, respectively[41]. We reasoned that combining these Cre-inducible *Mps1* knock-in (*CiMKi*) alleles (together and with wild-type (WT) *Mps1*) would result in an allelic series of CIN, ranging from very low (few missegregations with mostly mild errors) to very high (many missegregations with mostly severe errors).

*CiMKi* mice were born healthy and at Mendelian ratios. Activation of Cre recombinase by addition of 4-hydroxytamoxifen (4-OHT) to mouse embryonic fibroblasts (MEFs) from *CiMKi;Rosa26-CreER^{T2}* mice resulted in efficient recombination and expression of mutant *Mps1* mRNA (Supplementary Fig. 1D), from which mutant MPS1 protein was translated to comparable levels as wild-type protein (Supplementary Fig. 1E). As expected, the allelic series caused graded reductions in MPS1 activity, as

evidenced by acceleration of mitosis after mutant induction[42] (Fig. 1c) and reduced MAD1 levels at kinetochores[43] (Supplementary Fig. 1F). Time-lapse microscopy of 4-OHT-treated immortalized MEFs showed a striking increase in mitotic errors with diminishing MPS1 kinase activities (Fig. 1d, Supplementary Movies 1, 2), verifying the predicted phenotypes of the allelic series. As expected, induced CIN resulted in increased aneuploidy in primary MEFs (Supplementary Fig. 1G). Mutant induction also occurred efficiently in vivo in four-week old *CiMKi;Rosa26-CreER^{T2}* mice (Supplementary Fig. 1H), and analysis of anaphase figures in intestinal tissue sections showed that the expected range of CIN was induced (Fig. 1e). Moreover, single cell whole genome karyotype sequencing (scKaryo-seq[44]) showed that both aneuploidy and karyotype heterogeneity were enhanced in vivo in the small intestine one week after induction of moderate, high, and very high CIN (Fig. 1f, Supplementary Fig. 1I). We thus conclude that the CiMKi mouse model enables spatio-temporal control of a range of CIN in vivo.

### Moderate CIN leads to early intestinal tumor initiation.
Whole-body mutant inductions in *CiMKi;Rosa26-CreER^{T2}* mice disrupted small intestinal tissue organization (Fig. 2a). The extent of disorganization correlated with the degree of CIN and likely explained the severe weight loss seen in mice with high and very high CIN (Supplementary Fig. 2A). To study early CIN induction in the intestine without adverse effects on other organs, we generated *CiMKi;Villin-Cre* mice to enable mutant induction specifically in the intestinal tract from 12.5 days post coitum (dpc)[45]. All *CiMKi;Villin-Cre* mice were healthy and normally fertile, and showed no signs of intestinal dysfunction like diarrhea, weight loss (Supplementary Fig. 2B) or any other signs of health problems like abnormal posture, immobilization, or unresponsiveness. Also, we did not observe any abnormalities in general tissue characteristics at the age of 12 weeks or 8 months, except for the moderate CIN groups (Supplementary Fig. 2C, D). Moreover, moderate CIN had caused one or more lesions in the small intestine of these mice by as early as 12 weeks of age, as judged by methylene blue staining (Fig. 2b–d). Using histological analyses, we confirmed that these mice had indeed developed multiple low-grade adenomas (Fig. 2e, Supplementary Fig. 2C, D) of variable sizes. These adenomas were positive for nuclear β-catenin, showing that CIN was sufficient to induce constitutive Wnt pathway activation (Fig. 2f). Also, in this moderate CIN group, we detected a large low-grade adenoma in the colon of one mouse (Fig. 2g).

To get more insight in the differences in responsiveness between the various degrees of CIN on tumor initiation, we further analyzed colon and small intestinal tissues from the 12-weeks old mice for differences in survival and proliferative activity. We found no significant differences in the number of viable crypts (as determined by pHH3 positivity in each crypt; Supplementary Fig. 2E, F), nor in proliferative activities (ki67; Supplementary Fig. 2G, H). In both colon and small intestine, we observed an increasing trend with rising degrees of CIN for mitotic (pHH3 positive) cells per crypt (Supplementary Fig. 2I, J), and for apoptotic cells (based on morphology) per crypt (Supplementary Fig. 2K, L). However, we did not observe significant differences between the moderate to very high degrees of CIN.

CIN can thus lead to very early onset, spontaneous tumor initiation. This effect of CIN differed between the various degrees of CIN, with the strongest effect in mice with moderate CIN (Fig. 2c, d, g). Together, these data suggest that the various CIN levels differentially affect the chance of spontaneous intestinal tumor initiation. Examination of general tissue characteristics after

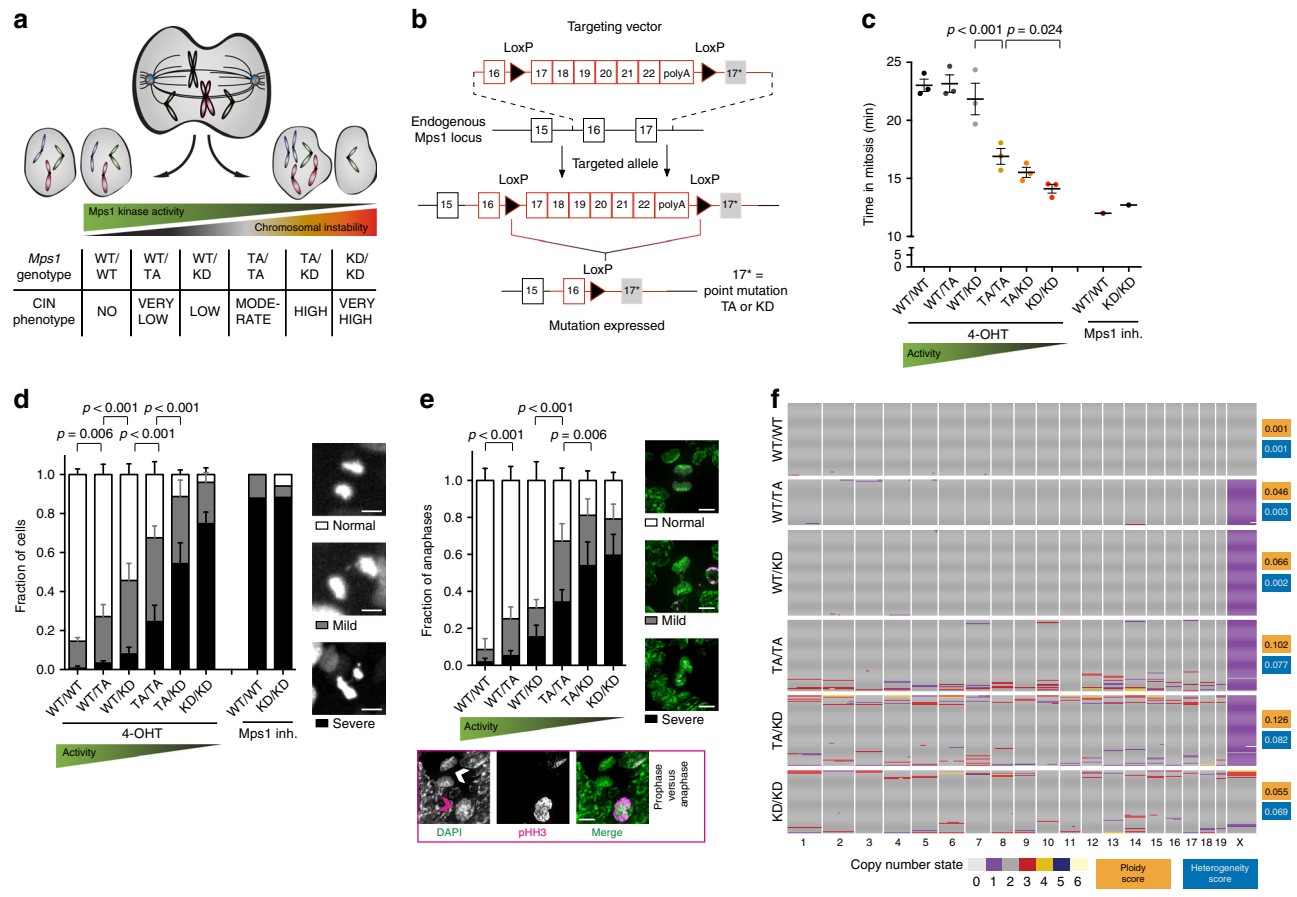

**Fig. 1 An allelic series for graded increases in CIN in vivo. a** Theoretical inverse correlation of decreased spindle assembly checkpoint function with increased severity of CIN. **b** Overview of the *CiMKi* alleles: the targeting vectors harbor a cDNA cassette of wild-type (WT) exons 17–22, flanked by lox-P sites, and the mutated exon 17* (TA or KD). In the targeted *CiMKi* alleles, wild-type *Mps1* is replaced with mutant *Mps1* upon Cre-mediated loxP recombination. **c** Quantification of time in mitosis (prophase to anaphase) by time lapse imaging of immortalized MEFs of the *CiMKi;Rosa26-CreER^T2* genotypes 56 h after 4-OHT addition. DNA was visualized by H2B-mNeon. Two independent untreated lines (*CiMKi^WT/WT* and *CiMKi^KD/KD*, both expressing wild-type *Mps1*) were treated with MPS1 inhibitor CPD5. Error bars indicate ± SEM of three independent MEF lines per genotype; 50 cell divisions per line. See also Supplementary Movies 1, 2. **d** Quantification of chromosome segregation fidelity of samples described in **c**. Missegregations of chromosomes were categorized as indicated: severe (≥three), mild (one or two). Scale bars 5 µm. See also Supplementary Movies 1, 2. Analysis as in **c**. **e** Chromosome segregation fidelity in situ in small intestine of *CiMKi;Rosa26-CreER^T2* mice one week after tamoxifen injection. DAPI (green) and anti-phospho-Histone H3 (Ser10) (pHH3; magenta) were used to identify anaphases (white arrowhead; weak staining) and prophases (magenta arrowhead; strong staining)), scale bars 5 µm. Graph shows quantification by category as in **d**, for at least 47 anaphases per small intestine. Bars indicate means ± SD (*n* = 12 (WT/WT), 8 (WT/TA), 11 (WT/KD), 7 (TA/TA), 5 (TA/KD), or 3 (KD/KD) mice per genotype; WT/WT group includes vehicle controls of other *CiMKi* genotypes). **f** scKaryo-seq (bin size 5 MB) showing ploidy in individual cells of small intestine 7 days after *CiMKi* induction. Graphs show individual cells (horizontal lines) of one example per genotype, see also Supplementary Fig. 1I. Average aneuploidy and heterogeneity scores are given for each genotype (*n* = 3). Colors indicate copy number state for a given chromosome. Statistics for panels **c**, **d**, and **e**: ordinary one-way ANOVA, uncorrected Fisher's LSD test, exact *p*-values are indicated when *p* < 0.05. Source data for panels **c**, **d**, and **e** are provided as a Source Data file.

CIN induction did not, however, provide an obvious explanation for this.

**Degree and site define oncogenic potential of CIN.** Human colorectal cancers (CRCs) are often aneuploid[46], and the vast majority of these cancers are caused by loss-of-function mutations in genes of Wnt pathway components such as *APC*[47–49]. Moreover, loss of heterozygosity (LOH) of *APC* causes extensive polyp growths in patients with familial adenomatous polyposis coli (FAP) syndrome[50–52]. To examine the impact of CIN on a tissue predisposed to cancer, we next investigated *CiMKi* mice carrying a mutant *Apc* allele (*Apc^Min/+*). *Apc^Min/+* mice normally develop around 30 adenomas in the small intestine and no or very few in the colon[53]. Note that the expected degrees of CIN in this tissue in *CiMKi* mice were directly verified by in situ analyses (see

Fig. 1e). In contrast to *CiMKi;Villin-Cre* mice, very high CIN in the *Apc^Min/+* background (*CiMKi^KD/KD;Apc^Min/+;Villin-Cre*) was embryonic lethal, precluding further analysis of this level of CIN. While mice with low CIN were sacrificed by the expected 12 weeks of age[30,36,54], mice with moderate or high CIN had to be sacrificed at 6–8 weeks due to severe weight loss. *Apc^Min/+* mice with moderate CIN presented with a striking increase in the amount of small intestinal adenomas (Fig. 3a–c, Supplementary Fig. 3A). Neither high nor low CIN, however, had the same effect, suggesting that adenoma formation in the small intestine is sensitive to a narrow range of CIN.

Macroscopic examination of the colons of the same *CiMKi; Apc^Min/+;Villin-Cre* mice revealed that in contrast to control and low CIN mice, colons from moderate and high CIN mice were widely covered with large adenomas (Fig. 3d–f, Supplementary

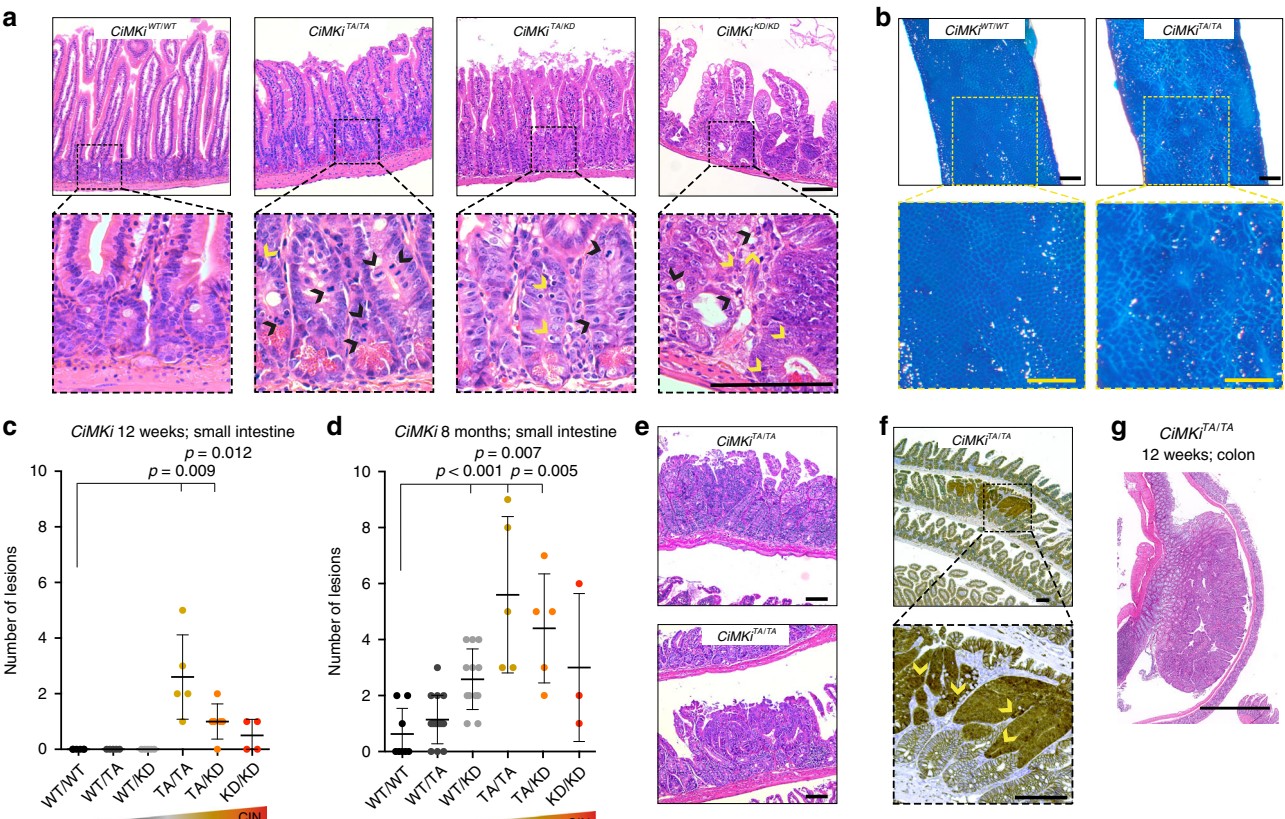

**Fig. 2 Moderate CIN causes early onset spontaneous tumor initiation in the intestine. a** H&E of *CiMKi;Rosa26-CreER^T2* small intestines one week after tamoxifen injection, showing aberrant crypt and cell size, hyperproliferation (black arrowheads indicate mitotic cells) and apoptotic bodies in crypt (yellow arrowheads), scale bars 100 μm. **b** Methylene blue stained, formalin-fixed whole mount small intestine of 12-week old *CiMKi;Villin-Cre* mice. Zoom boxes indicate normal (*CiMKi^WT/WT*) and aberrant (*CiMKi^TA/TA*) mucosa. Scale bars 1 mm. **c** Quantification of adenomas as determined on whole mount methylene blue staining in small intestine tissue in of 12-week-old *CiMKi;Villin-Cre* mice. Data represents mean ± SD, (*n* = 6 (WT/WT), 5 (WT/TA), 5 (WT/KD), 5 (TA/TA), 6 (TA/KD), or 4 (KD/KD) mice per genotype. **d** Quantification of adenomas as in **c**, but for small intestines of 8-month-old *CiMKi; Villin-Cre* mice (*n* = 8 (WT/WT), 14 (WT/TA), 12 (WT/KD), 5 (TA/TA), 5 (TA/KD), or 3 (KD/KD) mice per genotype. **e** H&E of small intestine adenomas from 12-week old *CiMKi^TA/TA;Villin-Cre* mice (moderate CIN), scale bars 100 μm. **f** ß-catenin immunohistochemistry on small intestine lesions in 12-week old *CiMKi^TA/TA;Villin-Cre* mice. Arrowheads indicate nuclear ß-catenin, scale bars 100 μm. **g** H&E staining of low-grade colon adenoma from a 12-week old *CiMKi^TA/TA;Villin-Cre* mice (moderate CIN). Scale bar 1 mm. Statistics for panels **c** and **d**: one-tailed Welch's *t*-test, comparing each group to *CiMKi^WT/WT; Villin-Cre*, exact *p*-values are indicated when *p* < 0.05. Source data for panels **c** and **d** are provided as a Source Data file.

Fig. 3B). This was most striking in the distal region of the colon, where aneuploid *APC*-mutant tumors also most frequently occur in humans[55,56]. Incidence was 100% (Supplementary Fig. 3C), and the number of adenomas was substantially higher than reported for other CIN models in the *Apc^Min/+* background[30,36,37]. Importantly, CIN in organoids established from these colon adenomas still corresponded to the expected levels (Fig. 3g, Supplementary Movies 3, 4), suggesting that high CIN levels were not selected against after adenoma formation and that there was no drift towards an 'optimal' CIN level. Of note: while individual adenoma sizes were comparable between all induced CIN levels (Supplementary Fig. 3D–G), adenomas with moderate or high CIN level had reached this size substantially earlier (6–8 vs. 12 weeks). We thus hypothesize that CIN advanced initiation, accelerated growth, or both.

In humans, tumors in the distal part of the colon are often considered CIN as they are typically aneuploid and karyotypically heterogeneous[20,57]. Since the *CiMKi;Apc^Min/+;Villin-Cre* mice with moderate and high CIN mimicked such distal colon tumors, we next assessed aneuploidy and heterogeneity of copy number alterations (CNAs) of colon adenomas. scKaryo-seq showed that both aneuploidy and karyotype heterogeneity were increased with

moderate and high CIN (Fig. 3g, h, Supplementary Fig. 3H). Chromosome 18, which harbors the *Apc* allele (that is often subject to LOH in human FAP tumors[47,53]), was diploid in the vast majority of cells. Since adenoma formation in *Apc^Min/+* mice requires LOH of wild-type *Apc*[58] and since *CiMKi* colon adenoma organoids grew independently of Wnt ligands, this indicated that LOH of *Apc* by CIN occurred in a manner other than whole chromosome 18 loss, as previously suggested[30]. Targeted PCR detected only *Apc^Min* alleles (Supplementary Fig. 3I), strongly suggesting that LOH was accomplished either by double non-disjunction events of both chromosomes 18 or by somatic recombination[30,59], the latter of which is likely the cause of *APC* LOH in FAP patients[51,60]. Both these processes could be accelerated by CIN.

**Enhanced proliferation in colon but not in small intestine.** Our data thus far show that the effect of CIN on karyotype heterogeneity and tumor formation in identical genetic backgrounds depends on the degree of CIN and the tissue in which CIN occurs. As high CIN caused massive colonic adenomas but did not increase adenoma formation in the small intestine, the effects of a similar range of CIN can be profoundly different

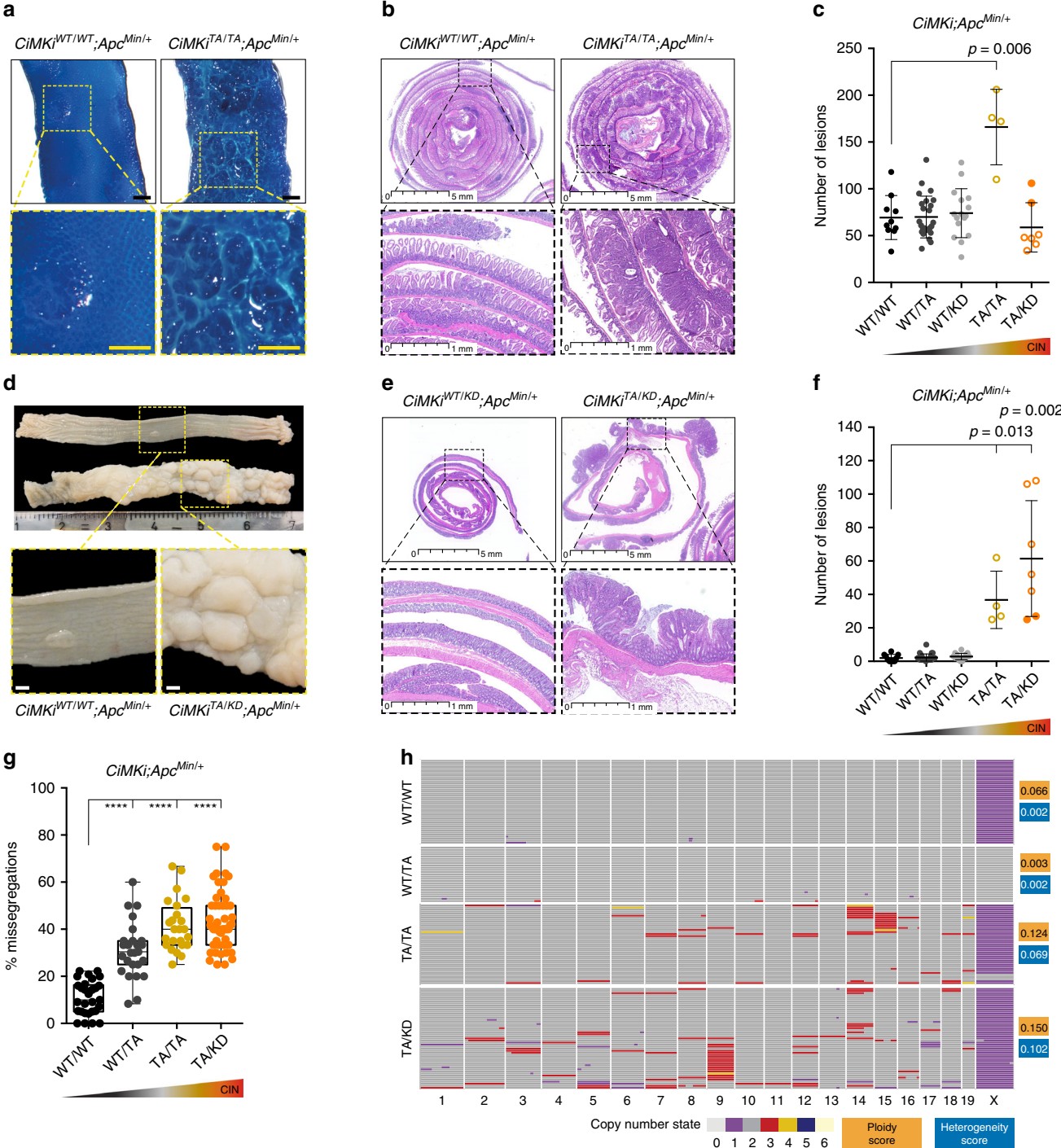

between tissues. To better understand the tissue-dependent sensitivities to CIN, we first assessed the possibility that the same *CiMKi* mutations had resulted in different CIN levels in small intestine vs. colon. Although time-lapse imaging of colon and small intestinal organoids showed that CIN levels were not identical between the two tissues (in $Apc^{Min}$ background), genotypes with drastically different impacts on tumor formation (Fig. 3c, f) had comparable CIN levels (e.g., $CiMKi^{TA/TA}$ in colon vs. $CiMKi^{TA/KD}$ in small intestine) (Fig. 4a, b). Strikingly however, moderate and high CIN caused significant expansion of the proliferative compartments in the colons of *CiMKi; $Apc^{Min/+}$;Villin-Cre* mice at four weeks of age (roughly the time

of adenoma initiation), but not in the small intestine (Fig. 4c–f, Supplementary Fig. 4A, B). The percentage of proliferating cells within the compartment (proliferative index) was similar across genotypes (Supplementary Fig. 4C, D), thus the total amount of proliferating cells in the crypts was enhanced (Fig. 4c, d, Supplementary Fig. 4E, F). Cells might therefore be more readily retained in a proliferate state in the colons of moderate and high CIN mice, increasing the chance that transformed cells propagate in colonic crypts. The fact that this increased proliferative state was not observed in the small intestine again underscores the difference in CIN response between these tissues.

**Fig. 3 Degree and site define oncogenic potential of CIN in tumor-prone intestines. a** Methylene blue stained whole mount small intestines of *CiMKi; Apc^Min/+;Villin-Cre* mice, showing mucosal architecture and abnormalities. Scale bars 1 mm. **b** H&E sections of small intestines of mice from panel **a**. **c** Quantification of small intestine adenomas from *CiMKi;Apc^Min/+;Villin-Cre* mice. Open dots represent mice euthanized at 6–8 weeks of age, closed dots represent mice at 12 weeks of age (*n* = 10 (WT/WT), 25 (WT/TA), 16 (WT/KD), 4 (TA/TA), or 7 (TA/KD) mice per genotype), data represents mean ± SD. **d** Formalin-fixed whole mount colons of *CiMKi;Apc^Min/+;Villin-Cre* mice, with adenomas predominantly located in the distal colon. Zooms indicate adenoma(s) in both genotypes. Scale bars 1 mm. **e** H&E sections of colons of mice from panel **d**. **f** Quantification of colon adenomas from *CiMKi;Apc^Min/+; Villin-Cre* mice. Representation as in **c**. **g** Quantification of chromosome segregation fidelity by time lapse imaging of colon adenoma organoid lines from different mice per genotype (*N* = 3 (WT/WT, WT/TA, TA/TA), or *N* = 5 (TA/KD). Percentages of missegregations per organoid were quantified (*n* = 29 (WT/WT), *n* = 27 (WT/TA), *n* = 24 (TA/TA), or *n* = 51 (TA/KD), from at least 5 divisions per organoid. Box-plot: each dot is one organoid, center line is median, box extends from 25th to 75th percentile, whiskers show min-to-max. **h** scKaryo-seq (bin size 5 MB) showing ploidy in individual cells of colon adenomas per genotype: aneuploidy and heterogeneity scores are given for each sample (*n* = 2 (TA/TA, TA/KD), or *n* = 3 (WT/WT, WT/TA), see also Supplementary Fig. 3H). Graphs show individual cells (horizontal lines) of one example per genotype. Colors indicate copy number state for a given chromosome. Statistics for panels **c**, **f**, and **g**: one-tailed Welch's *t*-test, comparing each group to *CiMKi^WT/WT;Apc^Min/+;Villin-Cre*, exact *p*-values are indicated when *p* < 0.05, and *p* < 0.0001 is indicated by asterisks (****). Source data for panels **c**, **f**, and **g** are provided as a Source Data file.

## Discussion

Aneuploidy and CIN are hallmarks of cancer, yet despite impressive efforts to model CIN in tumorigenesis, its importance to tumor development remained unclear. Our mammalian model for inducible, graded CIN levels, combined with direct visualization of chromosome segregation error frequencies in the relevant tissues allowed us to show that (1) moderate to high CIN levels are sufficient to drive adenoma formation at early age, (2) the maximum effect on tumor formation is achieved by distinct CIN levels, and (3) this differs between tissues with an identical cancer predisposition mutation.

Although previous studies have reported spontaneous tumor formation in CIN mice, it occurred sporadically and with late onsets of more than 12 months of age[21–29]. By contrast, our *CiMKi* mice with moderate CIN (*CiMKi^TA/TA*) developed a substantial number of lesions in the small intestine, and in one case in the colon, as early as 12 weeks of age. This shows that CIN is a more potent driver of tumor initiation than previously thought. As our model was able to probe the effects of a wide range of CIN, it is possible that the optimal CIN level for tumor induction was not reached in prior studies. The inducible nature of the CiMKi model is most probably a prerequisite for reaching the higher CIN levels. Many other CIN models were not inducible or tissue-specific, resulting in embryonic lethality in homozygous knock-outs, possibly due to severe missegregations during the developmental stages[22,23,61–63]. Also, we used the CiMKi model here to induce CIN locally in the gastro-intestinal tract, thereby preventing adverse effects in other tissues.

Despite the outstanding tumorigenic potential of specifically the moderate degree of CIN, we did not find significant differences in survival or proliferative activities of crypts, nor in apoptotic responses between the moderate and the higher degrees of CIN. This indicates that additional factors may play a role, and/or that the moderate degree of CIN is the most effective in inducing aneuploidies that are beneficial for adenoma initiation or growth. Importantly, we confirmed with scKaryo-seq that the higher degrees of CIN lead to aneuploid populations in healthy intestinal tissue. Aneuploidies were also observed in adenomas that arose after induction of moderate or high CIN, suggesting that at least a portion of the aneuploid populations induced by CIN are being propagated as proliferative cells in the adenomas.

Our model enabled us to directly compare the effects of the various CIN levels between two different tissues -small intestine and colon- within the same mice. Whereas in the *Apc^Min/+* background moderate CIN levels markedly increased adenoma burdens in both tissues, a higher CIN level did not affect the small intestine, but increased the adenoma burden in the colon even more. Our data therefore do not support the previously proposed

model in which low CIN levels promote tumorigenesis while high CIN leads to cell death and tumor suppression[32,64]. Instead, we argue that the role of different CIN levels is much more complicated, as similar levels of CIN can have contrasting effects in distinct tissues.

Several factors might account for the different effects between small intestine and colon: our finding that moderate and high CIN leads to an enhanced proliferative state in colon but not in small intestine shows a remarkable difference in response between the two tissues. Hyperproliferation of crypts could be an adaptive response to cell death or arrest due to high CIN. In that case, differences in tolerance for high CIN between the two tissues could play a role. Enhanced proliferation can increase the chance that transformed cells propagate in colonic crypts[65], however it does not account for the tumorigenic effect of moderate CIN in the small intestine. Another explanation might be differential impacts of *Apc* loss on CIN between colon and small intestine: it was previously reported that loss of *Apc* impacts on fidelity of chromosome segregation[66]. However, in a more recent study it was reported that inactivating *APC* mutations in human organoids do not significantly induce CIN[67]. Importantly, in our own experiments we found that the frequency of errors in *Apc^Min/+* organoids (*CiMKi* wild-type) is very low, and that the *Apc^Min/Min* tumors are diploid, suggesting that the *Apc* mutation alone does not induce substantial CIN. Therefore, other factors besides CIN itself might play a role as well, such as the normal variations throughout the gut in stem cell number and physiological Wnt activity[68] or in the adaptive immune landscape[69]. It will be exciting to further investigate the underlying mechanisms, as it can impact on future treatment strategies for different cancer types. The tissue-specific inducible nature of the CiMKi model enables studying the impact of various CIN levels in many other organs as well.

*Apc^Min/+* mice[54,70] have been widely used to model human FAP, the hereditable form of CRC[52]. Tumorigenesis in these mice requires LOH of *Apc*, which directly mimics the human disorder, and is genetically comparable to tumorigenesis in FAP patients. However, whereas in human FAP patients mostly colon tumors occur, *Apc^Min/+* mice develop many adenomas in the small intestine but very few in the colon. It is therefore intriguing that addition of moderate to high CIN to this model causes early occurrence of colon adenomas, thus making it a better model for human disease. Furthermore, as in human FAP, colonic adenomas of *CiMKi;Apc^Min/+* mice are predominantly located in the distal colon and are aneuploid and karyotypically heterogeneous[71]. Also, in contrast to adenomas from *Apc^Min/+* mice[58], LOH in *CiMKi;Apc^Min/+* mice did not occur by loss of (part of) the wild-type allele. Instead, we found disomy of the mutant chromosome 18 in the adenomas, similar to a CIN model driven

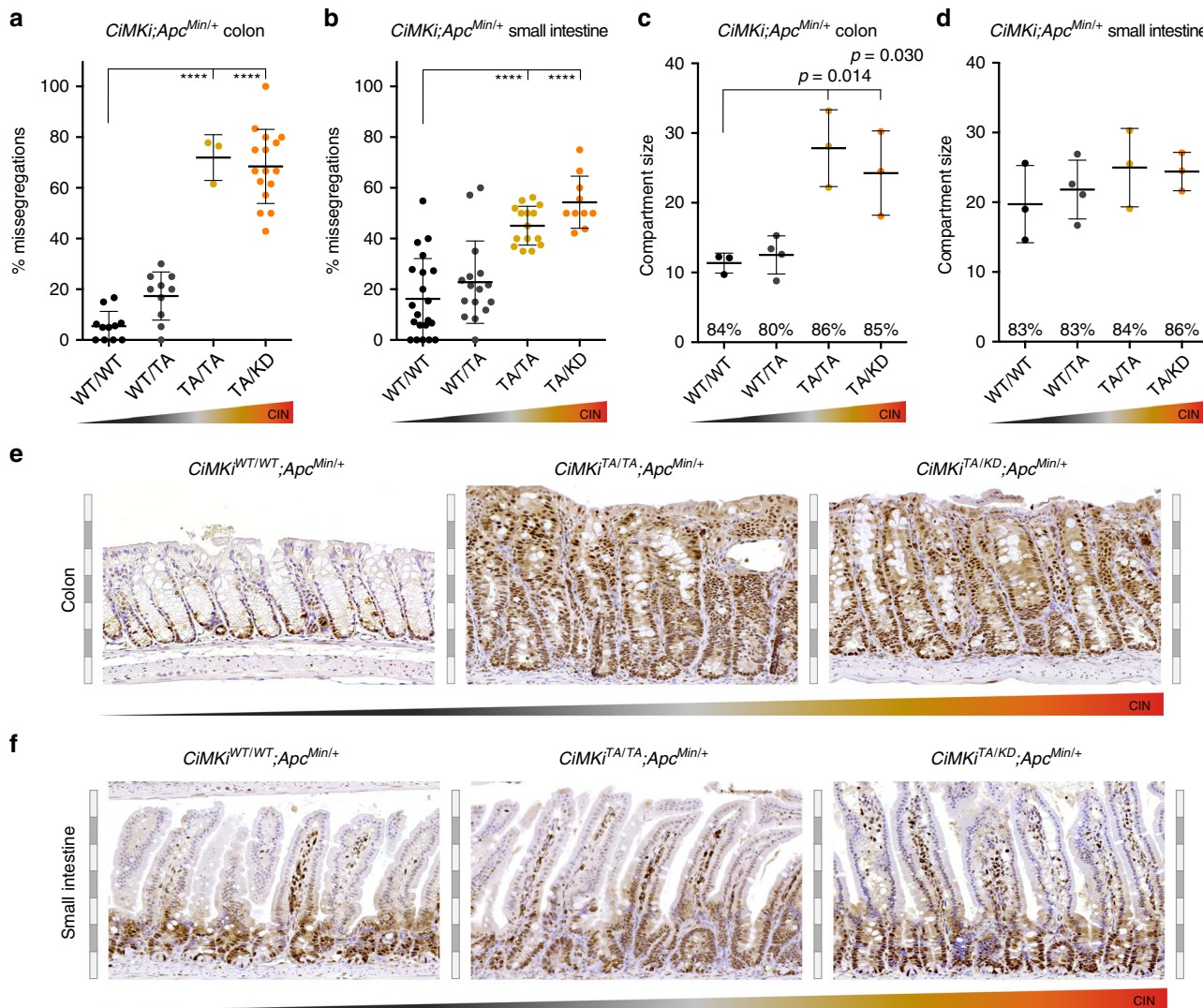

**Fig. 4 Enhanced proliferation in colon but not in small intestine. a**, **b** Quantification of chromosome segregation fidelity by time lapse imaging of colon organoids ($n = 11$ (WT/WT), $n = 10$ (WT/TA), $n = 3$ (TA/TA), $n = 16$ (TA/KD)) (**a**) and small intestine organoids ($n = 21$ (WT/WT), $n = 16$ (WT/TA), $n = 15$ (TA/TA), $n = 10$ (TA/KD)) (**b**), with at least 5 divisions per organoid, from various *CiMKi;Apc^Min/+;Villin-CreER^T2* genotypes. Data represents mean ± SD, each dot is one organoid. **c**, **d** Proliferative compartment in colon (**c**) and small intestine (**d**) of 4-week old *CiMKi;Apc^Min/+;Villin-Cre* mice as determined on ki67 stained tissue sections by scoring the number of cells between the first positive cell at the bottom of the crypt and the last positive cell in the transit amplifying zone (for example images see Supplementary Fig. 2G, H). Dot plots show the average size of the compartment for each mouse (10 crypts (with normal appearance, selected from similar regions (~2/3 from proximal site) per mouse), and mean ± SD of 3 mice (WT/WT, TA/TA, TA/KD), or 4 mice (WT/TA) per genotype. Percentages indicate proliferative index (percentage of ki67 positive cells within compartment) for each genotype. **e**, **f** PCNA staining of tissue sections from colon (**e**) and small intestine (**f**) of 4-week old *CiMKi;Apc^Min/+;Villin-Cre* mice, as a proxy for proliferative activity, showing increased size and PCNA positivity of colon crypts from *CiMKi^TA/TA;Apc^Min/+;Villin-Cre* and *CiMKi^TA/KD;Apc^Min/+;Villin-Cre* mice. Images are from one of three mice per genotype (see Supplementary Fig. 4A, B). Alternating gray and white scale bars 50 μm. Crypts were selected from similar regions (~2/3 from proximal site). Statistics for panels **a**–**d**: one-tailed Welch's *t*-test, comparing each group to *CiMKi^WT/WT;Apc^Min/+;Villin-Cre(ER^T2)*, exact *p*-values are indicated when $p < 0.05$, and $p < 0.0001$ is indicated by asterisks (****). Source data for panels **a**–**d** are provided as a Source Data file.

by Bub1 insufficiency[30]. Other processes by which LOH can be achieved are somatic recombination events[59], as was described for *APC* LOH in human FAP tumors[51,60], or double non-disjunction events during mitosis[30]. CIN can be involved in both processes: DNA damage and double strand breaks as a result of CIN may be repaired through recombination with the mutant allele, and doubling of the mutant chromosome accompanied or followed by loss of the wild-type allele by non-disjunctions during anaphase can lead to disomy of the mutant. So even though the exact mechanism remains to be uncovered, LOH of *Apc* in the *CiMKi; Apc^Min/+* mice is most probably accelerated by CIN. Importantly,

in the colon but not in the small intestine, moderate CIN had a very high tumorigenic potential in *Apc^Min/+* mice, but only very modest in the wild-type background. This could indicate that specifically in the colon, CIN promotes (by LOH) rather than initiates tumor formation, again underlining the differences in responses to CIN between the two tissues. Taken together, *CiMKi;Apc^Min/+* mice are a useful model to study sporadic and hereditary human CRC.

In conclusion, it is now possible with the CiMKi model to accurately study the interaction between CIN and tumor development in a host of tissues and genetic backgrounds. Because of

tight spatio-temporal control of CIN, CiMKi also enables investigations into the effect of various levels of CIN on cancer cell dissemination, as well as on possible tumor regression. The latter may greatly aid ongoing efforts that examine if exacerbating CIN, for example by MPS1 small molecule inhibitors, has potential as cancer therapy.

## Methods

**Mice: strains, experiments, and analysis**. All animal experiments were approved by Animal Experimental Committee and the Dutch Central Authority for Scientific Procedures on Animals (CCD). All animals were bred and housed under standard conditions (humidity 50–60%, 22–23 °C, inverted 12/12 h light/dark cycle, water and standard chow ad libitum) at the animal facility of the Gemeenschappelijk Dieren Laboratorium (GDL), Utrecht, the Netherlands, and the Hubrecht animal facility.

Genetically modified mice strains used in this study include: *ACTB:FLPe* (B6. Cg-Tg(ACTFLPe)9205Dym/J, stock number 005703), *Rosa26-CreER$^{T2}$* (B6.129Gt (ROSA)26Sor$^{tm1(cre/ERT2)Tyj}$/J, stock number 008463), and *Apc$^{Min/+}$* (C57BL/6J-Apc$^{Min}$/J, stock number 002020) and were purchased from JAX® mice. *Villin-Cre* mice we a gift from S. van Mil (originated from JAX® mice (B6.Cg-Tg(Vil1-cre) 997Gum/J, stock number 004586). *Villin-creER$^{T2}$* mice were a gift from J. van Rheenen. All mice were maintained in C57BL/6 background. Mice were genotyped using standard PCR and targeted sequencing procedures. For primers see Supplementary Table 1.

*CiMKi* mice were generated under license of UMCU (DEC 2010.I.02.026). The *CiMKi* alleles were designed following the example of the BRAF-V600E inducible model by Dankort et al.[72]: a cDNA cassette of wild-type exons 17–22 is followed by a stop codon and polyA sequence. The cassette is flanked by loxP sites, and the 3' recombination arm harbors one of the point mutations in exon 17 (D637A: GAT > GCT; T649A: GCA > ACA). The cassette was cloned into the pAC16 targeting vector (kind gift from J. Jonkers). For a detailed outline of construction of CiMKi vectors see Supplementary Methods. 129/Ola-derived IB10 ES cells (kind gift from H. Clevers) were electroporated (Biorad gene pulser) with the linearized targeting construct pAC16-D637A or pAC16-T649A. Targeted cells were selected with puromycin (1 µg ml$^{-1}$, Sigma) and single colonies were subsequently picked and cultured in 96-wells plates.

Individual clones were analyzed for presence of the *CiMKi* alleles using standard PCR (Primers used: Forward TCTATGGCTTCTGAGGCGG and Reverse AAGGGACATCAGGGAAGCAA). DNA from targeted ES cells yielded a band of ~2.8 kb.

Southern blot was performed according to standard protocols to confirm correct integration of the *CiMKi* alleles. 5' probe (500 bp) was obtained from genomic DNA from 129/Ola-derived IB10 ES cells, and labeled using a standard Rediprime II Random Prime labeling system (GE healthcare) and radioactive [α −32P] dCTP. Digestion of genomic DNA from ES cells with EcoRV and hybridization with the 5' probe resulted in a 9.5 kb band (when wildtype) or 4.7 kb band (when targeted with *CiMKi* allele).

Confirmed targeted ES cell clones for both *CiMKi* mutations were injected into C57BL/6 blastocyst, which were then transplanted into pseudo-pregnant females (standard techniques, performed under the license of the GDL Utrecht). Chimeric mice were bred with C57BL/6 mice to obtain germline transmission. Agouti mice were then backcrossed six times into a C57BL/6 background. Genotypic analysis of offspring was performed using standard PCR and targeted sequencing (Supplementary Table 1). To remove the puro cassette from the original pAC16 construct, *CiMKi* mice were bred with *ACT-Flp* mice (C57BL/6 background). Only lines that showed loss of the puro cassette (as confirmed by standard PCR) were used to maintain *CiMKi* lines.

To induce loxP recombination, MEFs were treated with 4-hydroxy-tamoxifen (4-OHT; 1 µM, Sigma H6278). Mice were injected intraperitoneally with Tamoxifen (1 mg dissolved in corn oil; Sigma, C8267). In *CiMKi;Apc$^{Min}$;Villin-Cre* mice, *CiMKi* alleles were induced at 12.5 dpc. when the Villin promotor is activated. To confirm recombination, RNA was isolated with a quick RNA kit (Zymo Research). cDNA was prepared using standard procedures, subjected to PCR and subsequently sequenced to determine the presence of T649A or D637A. For primers see Supplementary Table 1.

Mice were sacrificed at four weeks, twelve weeks or eight months of age, and immediately dissected. Small intestine was separated from colon, both were flushed with PBS and pieces of tissue were snap-frozen for later RNA/protein analysis. The organs were stored in formalin until further processing.

**Histology and immunohistochemistry**. Formalin fixed intestines were cut open longitudinally and stained with 0.25% methylene blue in dH$_2$0, and rinsed with PBS. Pictures were taken with ×6.3 magnification using an Olympus SZX stereo microscope to count the number of lesions in the small intestine and colon. After washing with PBS to remove the methylene blue, intestines were rolled into "Swiss rolls" for paraffin embedding.

For identification and assessment of lesions 4-µm sections of paraffin-embedded tissue were cut and stained with hematoxylin/eosin (H&E). These slides were scanned (Nanozoomer XR, Hamamatsu) for digital image analysis using NDP.view2 Software from Hamamatsu. Grading of dysplasias was done following the existing guidelines for human intestinal adenomas.

Apoptotic bodies were recognized on H&E stained sections, according to strict morphological criteria such as cell shrinkage with retracted pink to orange cytoplasm, chromatin condensation and nuclear fragmentation and separation of cells by a halo from adjacent enterocytes.

For proliferation measurements slides were incubated with ki67 antibody (ThermoFisher, RM-9106, ARS pH9, 1:50 and 3-h incubation), anti-Rabbit Envision-HRP (DAKO, 1 h) as secondary antibody, and counterstained with hematoxylin. ki67 positive cells were counted from the bottom to the top of the crypt till the upper most positive cell. The proliferative compartment was defined as the part of the crypt between the bottom and the upper most labeled cell. The proliferative activity (ki67 index) was calculated as the percentage of positively labeled cells divided by the total number of counted cells within the proliferative compartment. ß-catenin localization was assessed on paraffin sections stained with anti-ß-catenin (BD Transduction Laboratories, Clone 14/ Beta-Catenin 610154, 1:1000, overnight incubation), anti-Mouse Envision-HRP (DAKO, 1 h) as secondary antibody, and counterstained with hematoxylin. PCNA was stained with anti-PCNA (Clone PC10; Millipore MAB424, 1:100, overnight incubation), anti-Mouse Envision-HRP (DAKO, 1 h) as secondary antibody, and counterstained with hematoxylin. pHH3 was stained with anti-phospho-Histone H3 (Ser10) (Millipore 06-570), 1:1000, overnight incubation), anti-Rabbit Envision-HRP (DAKO, 1 h), as secondary antibody, and counterstained with hematoxylin.

**Isolation of MEFs**. *CiMKi* mice were bred with *Rosa26-CreER$^{T2}$* mice and maintained in a stable homozygous *CreER$^{T2}$* background. Pregnant females were sacrificed at 13–17 dpc. by cervical dislocation. Uterine horns were dissected out and placed in tubes containing PBS. Embryos were separated from their placenta and surrounding membranes. Red organs, brains and tail (for genotyping) were removed. Embryos were finely minced using razor blades and the remaining cells/ tissues were suspended in a tube containing 2 ml Trypsin and kept at 37 °C for 15 min. Two volumes of media (DMEM supplemented with 10% FBS, non-essential amino acids, glutamin and Pen/Strep) were added and remaining tissues were removed by allowing them to settle down at the bottom of the tube. Supernatant was subjected to centrifugation for 5 min at 1000 rpm, cell pellet was resuspended in medium and plated in 10 cm dishes.

**MEFs (immunofluorescence and live cell imaging)**. For immunofluorescence cells were plated on 12 mm coverslips and harvested after 1 h nocodazole (250 ng ml$^{-1}$, Sigma, M1404) and MG132 (2 µM, Sigma C2211) treatment. Cells were pre-extracted with 0.1% Triton X-100 in PEM (100 mM PIPES (pH 6.8), 1 mM MgCl$_2$ and 5 mM EGTA) for 1 min at 37 °C before fixation with 4% paraformaldehyde in PBS. Coverslips were subjected to antibody staining following standard procedures (primary antibodies anti-Mad1 (Santa Cruz sc67337, 1:1000), anti-Centromere Protein (ACA) (Antibodies Incorporated 15-234-0001, 1:2000). Images were acquired on a DeltaVision RT system (Applied Precision) with a ×100/1.40NA UPlanSApo objective (Olympus) using SoftWorx software. Images are maximum intensity projections of deconvolved stacks. Quantifications were done using ImageJ software and a macro[73] to threshold and specifically select kinetochores.

For live cell imaging, immortalized MEFs (transduced with large T and small T expressing lentivirus (Plasmid #22298, Addgene) were transduced with an H2B-Neon expressing lentivirus (pLV-H2B-Neon-ires-Puromycin)[67,74], and selected with puromycin (1 µg ml$^{-1}$, Sigma P7255). These stably H2B-mNeon expressing MEFs were plated in 24-well plates and imaged 56 h after 4-OHT treatment for 16 h in a heated chamber (37 °C and 5% CO2) using a ×20/0.5NA UPLFLN objective on an Olympus IX-81 microscope, controlled by Cell-M software (Olympus). Images were acquired using a Hamamatsu ORCA-ER camera and processed using Cell-M and ImageJ software.

**MEFs (mitotic spreads and Western blot)**. For mitotic spreads, MEFs were treated with STLC (1 µM, Sigma 164739) for 4 h. Mitotic cells (isolated by shake-off) were treated for 10 min in hypotonic buffer (75 mM KCl), fixed with acetic acid/methanol, dropped onto glass cover slides and stained with DAPI (1 mg ml$^{-1}$, Sigma 32670). Images were acquired on a DeltaVision RT system (Applied Precision) with a ×100/1.40NA UPlanSApo objective (Olympus) using SoftWorx software. Chromosomes were counted manually using Image J software.

For Western blot, MEFs were treated with 4-OHT or EtOH for 72 h, and then lysed with Laemmli buffer. Protein levels were assessed by standard Western blot procedures (anti-ESK (Santa Cruz sc-541, 1:1000), and anti-α-Tubulin (Sigma T5168; 1:10000)).

**Organoids (isolation, culture, and live microscopy)**. Organoids were isolated from healthy intestines of *CiMKi;Apc$^{Min/+}$;Villin-CreER$^{T2}$* mice or from colon adenomas. In brief, intestines of six-to-twelve weeks-old mice and

adenomas were dissected and cleaned with PBS. They were incubated in 0.5 mM EDTA on ice for 30 min (normal tissue), or EDTA treatment followed by 45 min in DMEM 2% FBS 1% Pen/Strep supplemented with 75U ml$^{-1}$ collagenase and 125 μg ml$^{-1}$ Dispase (adenoma tissue). Tissues were added to tubes with PBS, and crypts were removed from their niche by harsh shaking. After filtering the suspension using a 70 μm strainer, crypts or tumor cells were seeded in Matrigel (Corning, 356231). Organoids were cultured in medium containing advanced DMEM/F12 media (Invitrogen,126334-010), Hepes Buffer (Sigma, H0887, 1 mM), Pencilin/Strep (Sigma, P0781, 1%), Ala-Glu (Sigma, G8541, 0.2 mM), R-Spondin conditioned media (20%, kind gift from Hans Clevers) (wild-type only), Wnt conditioned media (50%, kind gift from Hans Clevers) (colon only), Noggin conditioned media (1%) (Thermo/Life Technologies, PHC1506, 1×), B27 (Thermo/Life Technologies, 17504001, 1×), nicotinamide (Sigma-Aldrich, 72340, 10 mM) (colon only), N-acetylcysteine (Sigma-Aldrich, A7250, 1.25 mM), EGF 0.1% (Invitrogen/Life Technologies, 53003-018) and Primocin 0.5% (Invivogen, ant-pm1). For passaging, organoids were sheared by repetitive pipetting and re-plated in Matrigel in a pre-warmed 24-well plate. For details see reference[75].

To establish stable organoid lines expressing H2B-mNeon, organoids were transduced with an H2B-Neon expressing lentivirus (pLV-H2B-Neon-ires-Blasticidin)[67,74], and selected with blasticidin (InvivoGen; 20 μg ml$^{-1}$). For induction of *CiMKi* alleles organoids were treated with 1 μM 4-OHT for 56 h. Organoids were seeded and imaged in 8-chamber IBIDI slides using a confocal spinning disk (Nikon/Andor CSU-W1 with Borealis illumination), equipped with atmospheric and temperature control. Organoids were imaged in XYZT-mode (12 to 20 z-sections at 2.5 μm intervals, for 8 to 12 h) at 37 °C at 3-min intervals, using a ×30 silicon objective and an additional ×1.5 lens in front of the CCD-camera. 3% 448 nm laser and 50 nm disk pinhole were used. Raw data were converted to videos using an ImageJ macro[67,76]. Fidelity of all observed chromosome segregations was scored manually, guided by the custom-made ImageJ macro for ordered data output.

**Single cell karyotype sequencing (scKaryo-seq).** Snap-frozen adenoma tissue was stained with 10 μg ml$^{-1}$ Hoechst 34580 (Sigma-Aldrich) and minced in a petri dish, on ice, using a cross-hatching motion with two scalpels. The minced tissue was kept on ice for 1 h after which it was filtered through 70 μm and 35 μm strainer. Nuclei were sorted in a 384-well plate containing 5 μl of mineral oil (Sigma) in each well and stored at −20 °C until further processing for library preparation and sequencing. For library preparation primers consisted of a 24 bp polyT stretch, a 4 bp random molecular barcode (UMI), a cell-specific 8 bp bar-code, the 5′ Illumina TruSeq small RNA kit adapter and a T7 promoter. mRNA of each cell was then reverse transcribed, converted to double-stranded cDNA, pooled and in vitro transcribed. Illumina sequencing libraries were prepared with the TruSeq small RNA primers (Illumina). Libraries were sequenced on an Illumina Nextseq 500 with 1 × 75 bp single-end sequencing[44,77]. The fastq files were mapped to GRCH38 using the Burrows-Wheeler Aligner. The mapped data was further analyzed using custom scripts in Python, which parsed for library barcodes, removed reads without a NlaIII sequence and removed PCR-duplicated reads. Copy number analysis was performed performed with the AneuFinder1.6.0 pipeline[78].

**Statistics and data reproducibility.** Power analysis predicted the number of animals that had to be used in each group to detect differences with 80% power and 95% confidence. Animals were not randomized, but assigned to the experimental groups based on their genotype. Statistical analyses were done with GraphPad Prism software. Comparisons between *CiMKi* wildtype and *CiMKi* mutants were made with a one-tailed Welch's *t*-test or with an ordinary one-way ANOVA, uncorrected Fisher's LSD test (stated in legends). Data is presented as mean ± SD unless otherwise stated in legends. All images and micrographs shown are representative for experimental groups of at least three individual mice in each experiment.

**Reporting summary.** Further information on research design is available in the Nature Research Reporting Summary linked to this article.

## Data availability

The scKaryo-seq data have been deposited in the European Nucleotide Archive database under the accession code PRJEB31573 [https://www.ebi.ac.uk/ena/data/view/PRJEB31573]. All the other data supporting the findings of this study are available within the article and its supplementary information files and from the corresponding author upon reasonable request. Source data for Fig. 1c–e; 2c, d; 3c, f, g; 4a–d; Supplementary Figs. 1a, b, e–g; 2a–l; 3a–g; 4c–f and a reporting summary for this article are available as Supplementary Information files.

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

## Acknowledgements

We thank H. Snippert for help with designing the *CiMKi* targeting vectors, J. Jonkers for advice and sharing reagents, and H. Clevers for sharing reagents and help with ES cell targeting. We thank S. van Mil and J. van Rheenen for donating mouse strains. We thank the Hubrecht mouse facility, the Hubrecht FACS facility, the Hubrecht Imaging facility, Single Cell Discoveries, and the Utrecht Sequencing Facility (USEQ) for assistance with the experiments. This work is part of the Cancer Genomics Centre and the Oncode Institute, and was further supported by the Dutch Cancer Society (grant numbers HUBR-2012-5321, HUBR-2012-5513, and 10126).

## Author contributions

W.H.M.H., A.J., R.H.M., G.J.P.L.K., and N.J. designed the research. W.H.M.H., A.J., and N.J. analyzed and assembled the data. A.I.Q. and B.G. assisted with animal maintenance and experiments. H.M., F.H.M.M., and G.J.A.O. performed pathology analyses. S.J.K. and N.L. performed and analyzed scKaryo-seq. A.T. assisted with organoid culture and experiments. W.H.M.H., G.J.P.L.K., and N.J. wrote the paper.

## Competing interests

The authors declare no competing interests.
