## [Peer Review File · Nature Communications]

Reviewers' comments:

Reviewer #1 (Remarks to the Author): Expertise in CIN

Chromosomal instability (CIN) leading to the loss and gains of chromosomes (aneuploidy) has long been proposed to be a potential driver of cancer development. Much debate has surrounded this topic largely due to contrasting reports arising from differences in tumor promotion vs. suppression, tissue type specificity, degree of CIN, fitness disadvantages to aneuploidy, etc. These controversies highlight the complexity surrounding this issue, including the lack of solid in vivo experimental models to test this hypothesis. Hoevenaar et al. sought to address this by generating cohorts of mice with mutations in the Mps1 kinase. These mice subsequently develop varying levels of CIN phenotypes, ranging from no CIN to very high CIN, and some exhibit early onset spontaneous tumorigenesis. The authors also create tissue-specific mutations in the small intestine and observed that CIN promotes adenoma formation. They further combined this in the APCmin background as a model for tumor predisposition. Although this study does not definitively solve the CIN/aneuploidy paradox, it is overall an excellent study developing and characterizing a great new model for producing graded CIN in vivo that warrants publication in Nature Communications. I only have a few minor comments and suggestions for improvement/clarification.

Small intestine adenoma formation more accurately reflects cancer initiation rather than the entire process of malignant transformation and therefore do not quite represent a full-blown model for tumorigenesis. The authors may want to make this point more explicit.

Figure 1C-D: The Mps1 inhibitor controls are important to establish the specificity of the Mps1 mutations in generating the observed phenotypes. Why was only one MEF line tested for this, and why only the KD/KD line? In addition, the authors should further clarify their definition of mild vs. severe errors beyond what is mentioned in the text.

Figure 1E: Anti-H3S10ph staining is poor and does not appear to reflect mitotic cells (at least in the images shown).

Figure 2D: TA/TA exhibits a striking increase in the number of lesions at 12 weeks, although WT/KD and TA/KD shows significantly less lesions. However, mitotic duration and mitotic error frequencies between these three genotypes are somewhat comparable. This effect was much less pronounced in 8-month-old animals. The authors should further discuss why the range of CIN permissable for adenoma formation is much narrower at early time points but broadens out at later time points.

Beginning with Line 193: The authors refute the model that low CIN promotes tumorigenesis but high CIN promotes cell death and tumor suppression, but then argue this point by saying it's complicated. The authors should explain and articulate their argument further to make their point clear.

Reviewer #2 (Remarks to the Author): Expertise in colorectal cancer in vivo

The manuscript by Hoevenaar and colleagues presents an account in which they generate a conditional mouse model of chromosomal instability by using knock in alleles of the spindle checkpoint kinase (Mps1) that they characterize in intestinal epithelium. Their conclusions are that a moderate degree of CIN alone is sufficient to generate adenomas in small intestine and in combination with heterozygous germline deficiency of the Apc tumor suppressor gene (Apcmin mice) can generate adenomas of the distal colon.

The study begins with a strong concept, good design and an appropriate initial characterization.

The authors succinctly present a review of an extensive and contradictory literature to justify their study. Regulating CIN and the extent to which it occurs via combinations of two Mps1 alleles of different efficacies (the mild T649A and more severe D637A mutations) seems an informed and intellectually sound decision. Similarly evaluating all 6 allelic combinations, including single and compound heterozygotes, nicely allows for different severities of CIN and therefore for identification of a sweet spot for generating resultant phenotypes. The initial characterization of the alleles in mouse embryo fibroblast cultures reasonably persuades that there is a graded phenotype immediately impacting on length of mitosis and chromosome segregation during anaphase.

Less clear from the characterization is the outcome of these mitotic events and which best allow for the propagation of aneuploidy within populations of cells or within tissues. The authors do consider this in the context of primary MEFs in FigS1G. The figure presents the distribution around 2n in a reasonable number of metaphase spreads per condition but the effects are modest and only recorded at a single early time point. How confident can the authors be that a given level of altered fidelity in chromosome segregation translates into aneuploid progeny that are still viable and proliferative in the tissue models presented?

This uncertainty continues in the analysis of the intestinal phenotypes. Consider Figure 2B-D: small intestinal adenomas are found with only one of the genotypes that confers a modest degree of CIN. This, with body weights in SFig2B, forms the sole analysis of the animals recruited to study. To claim only that animals were 'in health', based solely on body weight is insufficient. A much more comprehensive characterization of the intestinal epithelium in these animals is relevant, including survival analyses, gut pathology, detailed quantification of S-phase cells by BrDU incorporation, numbers of metaphases, and number of apoptotic cells. Such detailed profiling of both small intestine and colon would have allowed these to be compared and might have provided insight into their differences in responsiveness: what is the relationship between the fidelity of chromosome segregation in intestinal sections (as Fig1 E) and frequencies of apoptosis?

The analysis of adenomas with different degrees of CIN in Figure 2 is useful and persuades that the Mps1 T649A homozygous mutation induces early lesions that are characterized by elevated, nuclear beta-catenin. However, the analysis of the longer term 8 month old mice seems incomplete and not performed to the same depth or standard as for the younger mice or for the Apcmin model that follows. In particular were the wholemounts sectioned to look for microscopic lesions? What about the colon – was it analysed and were any lesions observed? This latter point is particularly relevant to the later claim that CIN induces colonic lesions on an Apcmin background, but is the effect to promote their growth or initiate them?

The Apcmin data documenting tumor formation towards the distal colon seems striking and the authors right to focus on it. The karyosequencing is informative as to how CIN induces loss/conversion of the APC wildtype allele and in showing ploidy changes in adenomas with the two allele combinations giving increased tumour burden.

The final results section on retention of aneuploid cells in small intestine versus colon seems uninformative. First, retention of cells would require some estimate of how migration was affected by CIN within both tissues and in both wildtype and Apcmin mice. The analysis they attempt with ki67 (and only in Apcmin mice) would be better performed with BrDU incorporation. This would allow both flash labelling to determine the number and distribution of S-phase cells, and with an appropriate chase period, to measure actual migration within the glands. Further analyses might also try and deal with cell cycle times in both tissues and how these are impacted by different severities of CIN.

In terms of the authors over-arching conclusion that the same level of CIN has different consequences in distinct tissues and contexts. This is probably correct but leaves open many alternative explanations (as the authors concede). The conclusion is slightly eroded by the extent to which checkpoints eliminate aneuploid cells and thus whether the degree of aneuploidy scales

with the degree of chromosomal instability; potentially crypts may adapt to a high level of cell ablation and become enlarged and hyperproliferative. These secondary consequences may also be tumour promoting.

Minor points

The authors decide to present results in main text based on severity of CIN but in figure legends use non intuitive allele names (TA and KD) that are not introduced or explained. This disconnect makes it hard to follow the flow of the paper, the authors should use allele names in main text: e.g. Mps1 as the gene name and allelic variants appropriately included as superscript.

FigureS2 (noting inappropriate figure title – the figure shows weight loss not tumour data) and associated text: Nowhere is it stated why the time course following whole body recombination only runs to 7 days. Presumably because mice certain allele combinations reached humane endpoint and had to be culled. This information should be included.

Nowhere is it considered that Apc loss itself may impact on CIN. (Kaplan et al. A role for the Adenomatous Polyposis Coli protein in chromosome segregation. *Nat Cell Biol* 2001;3(4):429–432.). This would seem potentially important. Is the frequency of aberrant chromosomal segregation increased with either heterozygous or homozygous Apc loss in colon or small intestine, potentially shifting the sweet spot for induction of aneuploidy?

Are the culture conditions for colonic and small intestinal organoids not different, extra Wnt is normally added to the former?

Regarding Figure 4, the authors should be careful to avoid regional differences in the height of the ki67+ compartment in different regions of the colon; they are dramatically different with more distal colon showing greater restriction towards the base.

Point-by-point response to remarks/suggestions from the referees

We were pleased to read that both reviewers appreciated our manuscript entitled ‘Degree and site of chromosomal instability define its oncogenic potential’, and would like to thank the reviewers for their useful comments and suggestions. These enabled us to greatly improve our manuscript. New experiments, text changes and more in-depth discussion are highlighted in yellow in the revised version of the manuscript. We answer and discuss all points raised by the reviewers in detail below (in blue).

Reviewer #1 (Remarks to the Author): Expertise in CIN

Small intestine adenoma formation more accurately reflects cancer initiation rather than the entire process of malignant transformation and therefore do not quite represent a full-blown model for tumorigenesis. The authors may want to make this point more explicit.

We agree and we made this point more explicit throughout the text, by replacing “tumorigenesis” with “tumor initiation” or “tumor formation”.

Figure 1C-D: The Mps1 inhibitor controls are important to establish the specificity of the Mps1 mutations in generating the observed phenotypes. We have used the Mps1 inhibitor also to show the effect of maximum reduction of MPS1 activity. The purpose hereof was to assess whether the “highest CIN level” mutant KD/KD can be compared with maximum inhibition by the inhibitor. This tells us that at the time of analysis (56 hours after induction), most if not all of the wild-type Mps1 protein is lost and replaced with mutant protein. We explained this now in the legend:

Revised legend: “Two independent untreated lines (WT/WT and KD/KD, both expressing wild-type Mps1) were treated with Mps1 inhibitor CPD5 to show the effects of maximal inhibition (comparable to induced KD/KD)”.

Why was only one MEF line tested for this, and why only the KD/KD line? We have used two MEF lines: WT/WT and KD/KD, both uninduced and thus expressing wild-type MPS1 protein. We realize that the text in the legend suggested differently. We have now clarified this in the figure itself and in the legend (see answer to previous comment).

In addition, the authors should further clarify their definition of mild vs. severe errors beyond what is mentioned in the text. We agree this was not clear and we have clarified this in the legend:

Revised legend: “Missegregations of chromosomes were categorized as indicated: severe (three or more), mild (one or two)”.

Figure 1E: Anti-H3S10ph staining is poor and does not appear to reflect mitotic cells (at least in the images shown). We have used H3Ser10ph staining to exclude prophase cells (high staining) from the analysis (to prevent false-positive scoring). Selecting mitotic cells with low (=poor) staining ensures inclusion of anaphases only (which lose their HH3 phosphorylation mark). We have now included an example of the staining in the different phases in the figure and explained in the legend:

Revised legend: [DAPI (green) and anti-H3S10ph (magenta) were used to identify] “anaphases (see examples on lower panel of prophase (magenta arrowhead; strong staining) and anaphase (white arrowhead; weak staining), scale bars 5 μ m.”

Figure 2D: TA/TA exhibits a striking increase in the number of lesions at 12 weeks, although WT/KD and TA/KD shows significantly less lesions. However, mitotic duration and mitotic error frequencies between these three genotypes are somewhat comparable. We respectfully disagree: we show the opposite (significant differences in mitotic duration and mitotic error frequencies between these three genotypes), which for us is an important argument for our overall conclusions. We do realize however that we didn’t include statistics on these data. We now have included statistics for Figures 1C (mitotic timing MEFs), 1D (errors MEFs), and 1E (errors in situ small intestine) that show the significant differences between the various groups to strengthen this point (please note that although there is not a significant difference between TA/TA and TA/KD in the mitotic timing in MEFs, the differences in the errors between these groups are significant in both MEFs and small intestine). Also, the exact numbers for these graphs are given in the Source Data file.

This effect was much less pronounced in 8-month-old animals. The authors should further discuss why the range of CIN permissible for adenoma formation is much narrower at early time points but broadens out at later time points. The range of CIN permissible for adenoma formation seems indeed narrower at 12 weeks. However, this range probably doesn’t permit adenoma formation in a black-or-white fashion, but rather increases the chance of a dividing cell to contribute to tumor initiation. This would explain the higher number of lesions at 8 months not only in the WT/KD and TA/KD animals but also in the TA/TA animals. We have clarified this in the results section:

Revised Results: “Moreover, we observed significant different effects between the various degrees of CIN, with the strongest effect in mice with moderate CIN (Fig. 2E, F, G). Together, it is likely that the various CIN levels differentially increase the chance of intestinal spontaneous tumor initiation.”

Beginning with Line 193: The authors refute the model that low CIN promotes tumorigenesis but high CIN promotes cell death and tumor suppression, but then argue this point by saying it's complicated. The authors should explain and articulate their argument further to make their point clear. We have further clarified our argument in the text (and also included additional suggestions by another reviewer):

Revised discussion: *“Instead, we argue that the role of different CIN levels is much more complicated, as similar levels of CIN can have contrasting effects in distinct tissues.*

Several factors might account for the different effects between small intestine and colon: our finding that moderate and high CIN leads to an enhanced proliferative state in colon but not in small intestine shows a remarkable difference in response between the two tissues. Hyperproliferation of crypts could be an adaptive response to cell death or arrest due to high CIN. In that case, differences in tolerance for high CIN between the two tissues could play a role. Enhanced proliferation can increase the chance that transformed cells propagate in colonic crypts⁶⁷, however it does not account for the tumorigenic effect of moderate CIN in the small intestine. Another explanation might be differential impacts of Apc loss on CIN between colon and small intestine: it was previously reported that loss of Apc impacts on fidelity of chromosome segregation⁶⁸. However, in a more recent report it was reported that inactivating APC mutations in human organoids do not significantly induce CIN⁶⁹. Importantly, in our own experiments we found that the frequency of errors in Apc^{Min/+} organoids (CiMKi wild-type) is very low, and that the Apc^{Min/Min} tumors are diploid, suggesting that the Apc mutation alone does not induce significant CIN levels. Therefore, other factors besides CIN itself might play a role as well, such as the normal variations throughout the gut in stem cell number and physiological Wnt activity⁶⁸ or in the adaptive immune landscape⁶⁹.”

Reviewer #2 (Remarks to the Author): Expertise in colorectal cancer in vivo

Less clear from the characterization is the outcome of these mitotic events and which best allow for the propagation of aneuploidy within populations of cells or within tissues. The authors do consider this in the context of primary MEFs in FigS1G. The figure presents the distribution around 2n in a reasonable number of metaphase spreads per condition but the effects *are modest and only recorded at a single early time point*. That is true, however, as (wild-type) MEFs have the tendency to become polyploid and aneuploid within few passages (in our hands around the 2nd or 3rd passage), we chose to analyze only cells at a very early timepoint. In our estimation, later timepoints would not give a reliable result.

*How confident can the authors be that a given level of altered fidelity in chromosome segregation translates into aneuploid progeny that are still viable and proliferative in the tissue models presented? Colon adenomas from the CiMKi; APC^{Min/+} mice have increased levels of aneuploidy and heterogeneity when moderate or high CIN levels were induced, compared to no CIN induction (Fig. 3H). This shows a correlation between CIN induction and aneuploidies in the resulting tumor. However, we do not claim or wish to claim that (different levels of) aneuploidies in the cancer cells underlie the various effects on tumorigenesis that we see; our data show that CIN induction in the intestine causes tumor phenotypes, but they do not speak to the underlying mechanism. We discuss this in the discussion section. Moreover, the ongoing CIN in our model will constantly create new aneuploid populations. This excludes faithfully analyzing aneuploidies over time (*in vitro* and *in vivo*), and attributing tumorigenic properties to specific aneuploid populations.*

We nevertheless agree that it is important to show, in addition to the *in vitro* MEF data, that the induced CIN indeed leads to aneuploidies *in vivo*. We therefore performed scKaryo-seq on intestinal tissues that after a short period (one week) of 4-OHT treatment. The results show that induction of moderate, high and very high CIN in the small intestine causes aneuploid progeny (Fig. 1F and S1I). The fact that CIN induction quickly generates aneuploid cells, and that aneuploidies are apparent in the end-stage tumors (Fig. 3H) suggests that viable aneuploid progeny indeed arise after CIN induction, and that it is likely that a subset of aneuploid cells are propagated as proliferative cells into the adenomas. We have added these data in Figs. 1F and S1I, and have adjusted the text accordingly in Results and Discussion sections:

Revised Results: *“Moreover, single cell whole genome karyotype sequencing (scKaryo-seq⁴⁴) showed that both aneuploidy and karyotype heterogeneity were enhanced *in vivo* in the small intestine one week after induction of moderate, high, and very high CIN (Fig. 1F, Supplementary Figure 1I).”*

Revised Discussion: *“Importantly, we confirmed with scKaryo-seq that the higher degrees of CIN lead to aneuploid populations in healthy intestinal tissue. Aneuploidies were also observed in adenomas that arose after induction of moderate or high CIN, suggesting that at least a portion of the aneuploid populations induced by CIN are being propagated as proliferative cells into the adenomas.”*

This uncertainty continues in the analysis of the intestinal phenotypes. Consider Figure 2B-D: small intestinal adenomas are found with only one of the genotypes that confers a modest degree of CIN. This, with body weights in SFig2B, forms the sole analysis of the animals recruited to study. *To claim only that animals were ‘in health’, based solely on body weight is insufficient. We agree that our description of “healthy” was poor, and we have now given a better description of the general health status of these animals:*

Revised Results: *“All CiMKi; Villin-Cre mice were healthy and normally fertile, and showed no signs of intestinal dysfunction like diarrhea, weight loss (Supplementary Figure 2B), or any other signs of health problems like abnormal posture, immobilization, or unresponsiveness.”*

A much more comprehensive characterization of the intestinal epithelium in these animals is relevant, including survival analyses, gut pathology, detailed quantification of S-phase cells by BrDU incorporation, numbers of metaphases, and number of apoptotic cells. Such detailed profiling of both small intestine and

colon would have allowed these to be compared and might have provided insight into their differences in responsiveness: what is the relationship between the fidelity of chromosome segregation in intestinal sections (as Fig1 E) and frequencies of apoptosis? We agree with the reviewer that a more comprehensive characterization of the intestinal epithelium of these animals is relevant as it might provide more insight in the differences in responsiveness between the various degrees of CIN. Rather than using more mice in a new BrdU or EdU pulse experiment, we opted to stain, as a proxy for proliferative activity, existing tissue sections for Ki67 and pHH3 to assess proliferative compartment and actively dividing cells, respectively. Altogether, we now give a detailed description for the gut pathology (also for the 8-month group; see next comment), analyzed survival and proliferative activity of the crypts, the number of mitotic cells and the number of apoptotic cells.

Revised Results: “Also, we did not observe any abnormalities in general tissue characteristics at the age of 12 weeks or 8 months, except for the moderate CIN groups (Supplementary Figure 2C, D). Moreover, moderate CIN had caused one or more lesions in the small intestine of these mice by as early as 12 weeks of age, as judged by methylene blue staining (Fig. 2B, E, G). Using histological analyses, we confirmed that these mice had indeed developed multiple low-grade adenomas (Fig. 2C, Supplementary Figure 2C, D) of variable sizes. These adenomas were positive for nuclear β -catenin, showing that CIN was sufficient to induce constitutive Wnt pathway activation (Fig. 2D). Also, in this moderate CIN group, we detected a large low-grade adenoma in the colon of one mouse (Fig. 2G).

To get more insight in the differences in responsiveness between the various degrees of CIN on tumor initiation, we further characterized analyzed colon and small intestinal tissues from the 12-weeks old mice for differences in survival and proliferative activity. We found no significant differences in the number of viable crypts (as determined by pHH3 positivity in each crypt; Supplementary Figure 2E, F), nor in proliferative activities (ki67; Supplementary Figure 2G, H). In both colon and small intestine, we observed an increasing trend with rising degrees of CIN for mitotic (pHH3 positive) cells per crypt (Supplementary Figure 2I, J), and for apoptotic cells (based on morphology) per crypt (Supplementary Figure 2K, L). However, we did not observe significant differences between the moderate to very high degrees of CIN.

To our knowledge, this is the first report of early onset, spontaneous tumor initiation as a result of CIN. This effect of CIN differed between the various degrees of CIN, with the strongest effect in mice with moderate CIN (Fig. 2E, F, G). Together, these data suggest that the various CIN levels differentially affect the chance of spontaneous intestinal tumor initiation. Examination of general tissue characteristics after CIN induction did not, however, provide an obvious explanation for this.”

Revised legend Supplementary Figure 2C, D: “In all groups, except for the moderate CIN groups, we observed normal mucosa of the small intestine and colon, and normal crypt villus ratio (1:3), brush border and epithelium lining consisting of goblet cells and enterocytes in the small intestine. Cellularity of the lamina propria was within normal limits, and scattered lymphoid aggregates were observed. In the colon, the crypts were regularly arranged and lined with goblet cells. The lamina propria contained normal cellularity, with scattered lymphoid aggregates.”

These analyses thus showed that these features cannot unequivocally explain the differential effects of the various degrees of CIN, and that other (secondary) factors might be involved, and/or that the moderate degree of CIN is the most effective to induce aneuploidies that are beneficial for adenoma initiation or growth. We now discuss this in the discussion section:

Revised Discussion: “Despite the outstanding tumorigenic potential of specifically the moderate degree of CIN, we did not find significant differences in survival or proliferative activities of crypts, nor in apoptotic responses between the moderate and the higher degrees of CIN. This indicates that additional factors may play a role, and/or that the moderate degree of CIN is the most effective to induce aneuploidies that are beneficial for adenoma initiation or growth.”

The analysis of adenomas with different degrees of CIN in Figure 2 is useful and persuades that the Mps1 T649A homozygous mutation induces early lesions that are characterized by elevated, nuclear beta-catenin. However, the analysis of the longer term 8 month old mice seems incomplete and not performed to the same depth or standard as for the younger mice or for the Apcmin model that follows. We now give a detailed a detailed description for the gut pathology for this group (see answer to previous question).

In particular were the wholemounts sectioned to look for microscopic lesions? We have now scored H&E sections from the whole mounts for the 8-month group and scored the lesions on the wholemounts from the 12- week old group. Whole mount counts for both groups are now both shown in the main figure (Fig. 2E, F), and the H&E counts are shown in Supplementary Figure 2C, D.

What about the colon – was it analysed and were any lesions observed? Together with the more thorough gut pathology that we have now performed for the 12 weeks old mice and for the 8-month old mice, we have now also scored for adenomas on these H&E stained sections. We found one big colon (low grade) adenoma in one mouse in the 12-weeks *CiMK^{TA/TA}* group. This is an important observation and we thank the reviewer for the suggestion. We have added this to the text and we have added an H&E staining for this tumor to the Figures (Fig. 2G). No colon adenomas were detected in any of the mice of the 8 months group.

Revised Results: “Also, in this moderate CIN group, we detected one large low-grade adenoma in the colon of one mouse (Fig. 2G).”

Revised Discussion: “[developed a substantial number of lesions in the small intestine], and in one case in the colon[, as early as 12 weeks of age]”

This latter point is particularly relevant to the later claim that CIN induces colonic lesions on an *Apcmin* background, but is the effect to promote their growth or initiate them? We agree and have discussed this point now in the discussion section.

Revised discussion: “Importantly, in the colon but not in the small intestine, moderate CIN had a very high tumorigenic potential in *Apc^{Min/+}* mice, but only very modest in the wild-type background. This could indicate that specifically in the colon, CIN promotes (by LOH) rather than initiates tumor formation, again underlining the differences in responses to CIN between the two tissues.”

The *Apcmin* data documenting tumor formation towards the distal colon seems striking and the authors right to focus on it. The karyosequencing is informative as to how CIN induces loss/conversion of the APC wildtype allele and in showing ploidy changes in adenomas with the two allele combinations giving increased tumour burden.

The final results section on retention of aneuploid cells in small intestine versus colon seems uninformative. First, retention of cells would require some estimate of how migration was affected by CIN within both tissues and in both wildtype and *Apcmin* mice. *The analysis they attempt with ki67 (and only in Apcmin mice) would be better performed with BrdU incorporation. This would allow both flash labelling to determine the number and distribution of S-phase cells, and with an appropriate chase period, to measure actual migration within the glands. Further analyses might also try and deal with cell cycle times in both tissues and how these are impacted by different severities of CIN. We agree that the ki67 analysis does not provide evidence for differences in retention of aneuploid cells. However, the only point we wish to make with these analyses is that proliferation rates are higher in the colon of high CIN mice. Although the mechanism is unclear, the Ki67 analysis supports an important message of our manuscript: that the small intestine and colon have intrinsically different responses to CIN. The proposed experiments are interesting but, in our opinion, do not outweigh the effort needed to perform them, for two main reasons. First, BrdU labeling to measure the effect of CIN on migration in the different tissues would require pulse-labeling of specifically the aneuploid cells, or the ability to distinguish BrdU-labeled aneuploid from euploid cells, which are not possible to the best of our knowledge. And second, since the proposed analyses on BrdU incorporation would not be able to identify whether migration of specifically the aneuploid cells is affected, we would not be in favor of using additional experimental animals to perform the BrdU pulse experiment (as mice in the original experiment did not receive BrdU or EdU).*

We nevertheless realize that our phrasing in the final results section might be an overstatement or at least confusing. To clarify, and in agreement with the Reviewer’s concerns on this point, we have reassessed our conclusion and have rewritten this paragraph accordingly. In addition, we have included PCNA stainings as an additional marker for proliferative cells to confirm the ki67 data (Fig. 4E, F, Supplementary Fig. 4A, B).

Revised Results (Title subheading & Title Fig. 4 & Supplementary Fig. 4): “Enhanced proliferation in colon but not in small intestine”

Revised Results (Conclusion paragraph): “Strikingly however, moderate and high CIN caused significant expansion of the proliferative compartments in the colons of *CiMKi;Apc^{Min/+};Villin-Cre* mice at four weeks of age (roughly the time of adenoma initiation), but not in the small intestine (Fig. 4C-F, Supplementary Figures 4A, B). As the percentage of proliferating cells within the compartment (proliferative index) was similar across genotypes, and thus the total amount of proliferating cells in the crypts was enhanced (Fig. 4C, D, Supplementary Figures 4C-F), cells might be more readily retained in a proliferate state in the colons of moderate and high CIN mice, increasing the chance that transformed cells propagate in colonic crypts. The fact that this increased proliferative state was not observed in the small intestine again underscores the difference in CIN response between these tissues.”

In terms of the authors over-arching conclusion that the same level of CIN has different consequences in distinct tissues and contexts. This is probably correct but leaves open many alternative explanations (as the authors concede). *The conclusion is slightly eroded by the extent to which checkpoints eliminate aneuploid cells and thus whether the degree of aneuploidy scales with the degree of chromosomal instability; potentially crypts may adapt to a high level of cell ablation and become enlarged and hyperproliferative. These secondary consequences may also be tumour promoting. The reviewer is right, and we have now discussed the possibility that such an adaptive response might differ between the two tissues, possibly due to differences in tolerance of cells to CIN.*

Revised discussion: “Hyperproliferation of crypts could be an adaptive response to cell death or arrest due to high CIN. In that case, differences in tolerance for high CIN between the two tissues could play a role.”

Minor points

The authors decide to present results in main text based on severity of CIN but in figure legends use non intuitive allele names (TA and KD) that are not introduced or explained. This disconnect makes it hard to follow the flow of the paper, *the authors should use allele names in main text: e.g. Mps1 as the gene name and allelic variants appropriately included as superscript. We have changed this accordingly: now we use the allele name (CiMKi) and its variants as superscripts as proposed here by the Reviewer throughout the text, and introduce the allele names TA and KD in the first paragraph of the results section for clarification:*

Revised Results: “we created mouse strains carrying a conditional T649A (TA) or D637A (KD; kinase-dead) mutation in the spindle assembly checkpoint kinase *Mps1*.”

FigureS2 (noting inappropriate figure title – the figure shows weight loss not tumour data) and associated text: Nowhere is it stated why the time course following whole body recombination only runs to 7 days. Presumably because mice certain allele combinations reached humane endpoint and had to be culled. This information should be included. We apologize for this mistake. This figure is now extended with more subfigures that do show tumor data, as well as analyses for survival and proliferative activity of the crypts, the number of mitotic cells and the number of apoptotic cells. We have changed the title accordingly, and added to the text that the mice indeed reached endpoint 7 days after whole body induction.

Revised legend title: “*Effects of CIN induction in the intestine*”

Revised legend: “At day 7, mice from the *CiMKi^{TA/KD}* and *CiMKi^{KD/KD}* groups reached the humane endpoint and were euthanized.”

Nowhere is it considered that *Apc* loss itself may impact on CIN. (Kaplan et al. A role for the Adenomatous Polyposis Coli protein in chromosome segregation. *Nat Cell Biol* 2001;3(4):429–432.). This would seem potentially important. Is the frequency of aberrant chromosomal segregation increased with either heterozygous or homozygous *Apc* loss in colon or small intestine, potentially shifting the sweet spot for induction of aneuploidy? In our experiments, we found that the frequency of errors in *Apc^{Min/+}* organoids (*CiMKi* wild-type) is very low, and that the *Apc^{Min/Min}* tumors are diploid (scKaryo-seq data), suggesting that the APC mutation alone do not induce CIN. This agrees with other recent data showing that inactivating APC mutations in human organoids do not lead to CIN (Drost et al., *Nature* 521, 43–47 (2015)). We have now referred to the Kaplan report and the Drost report and discussed this in the light of our own data.

Revised Discussion: “Another explanation might be differential impacts of *Apc* loss on CIN between colon and small intestine: it was previously reported that loss of *Apc* impacts on fidelity of chromosome segregation in cell lines⁶⁸. However, inactivating APC mutations in human organoids do not significantly induce CIN⁶⁹. Importantly, in our own experiments we found that the frequency of errors in *Apc^{Min/+}* organoids (*CiMKi* wild-type) is very low, and that the *Apc^{Min/Min}* tumors are diploid, suggesting that the *Apc* mutation alone do not induce significant CIN levels.”

Are the culture conditions for colonic and small intestinal organoids not different, extra *Wnt* is normally added to the former?

Colonic organoids indeed do not grow without addition of *Wnt*. We realize this was not stated in the method section and we have adjusted accordingly.

Adjusted Methods: “, *Wnt* conditioned media (50%, kind gift from Hans Clevers) (colon only),”

Regarding Figure 4, the authors should be careful to avoid regional differences in the height of the *ki67+* compartment in different regions of the colon; they are dramatically different with more distal colon showing greater restriction towards the base.

We realize there are differences between different regions, and have therefore selected similar regions (2/3 from proximal site) between animals. We have clarified this in the legend.

Adjusted Legend: “Dot plot shows the average size of the compartment for each mouse ($n=3-4$ per genotype, 10 crypts (with normal appearance, selected from similar regions (~2/3 from proximal site) per mouse)”

REVIEWERS' COMMENTS:

Reviewer #1 (Remarks to the Author):

I have now seen the revised version of this article and can recommend publication of it in Nature Communications. The authors have done a great job in addressing the referee comments. I believe this study will be an important contribution to the chromosomal instability and aneuploidy field.

Reviewer #2 (Remarks to the Author):

In their resubmission the authors have addressed all the initial criticism from their reviewers. The manuscript is significantly improved as a result and I have no hesitation in recommending it for publication in Nature Communications. In sum the paper identifies a useful experimental strategy to induce chromosomal instability (an important driver of the cancer process that is massively under represented by experimental models) into the intestinal tract. They then use it to identify sweet spots in the level of CIN that support the development of protumorigenic lesions in different regions of the bowel.